# Efficacy of a Rehabilitation Program Using Mirror Therapy and Cognitive Therapeutic Exercise on Upper Limb Functionality in Patients with Acute Stroke

**DOI:** 10.3390/healthcare12050569

**Published:** 2024-02-29

**Authors:** Jessica Fernández-Solana, Sergio Álvarez-Pardo, Adrián Moreno-Villanueva, Mirian Santamaría-Peláez, Jerónimo J. González-Bernal, Rodrigo Vélez-Santamaría, Josefa González-Santos

**Affiliations:** 1Department of Health Sciences, University of Burgos, 09001 Burgos, Spain; jfsolana@ubu.es (J.F.-S.); mspelaez@ubu.es (M.S.-P.); jejavier@ubu.es (J.J.G.-B.); rvs0014@alu.ubu.es (R.V.-S.); mjgonzalez@ubu.es (J.G.-S.); 2Faculty of Health Science, University Isabel I, 09003 Burgos, Spain; adrian.moreno@ui1.es; 3BioVetMed & SportSci Research Group, Department of Physical activity and Sport, Faculty of Sport Sciences, University of Murcia, San Javier, 30720 Murcia, Spain

**Keywords:** acute stroke, mirror therapy, cognitive therapeutic exercise, task-oriented training, upper limb, functionality, acute stroke, non-pharmacological therapy

## Abstract

Applying evidence-based therapies in stroke rehabilitation plays a crucial role in this process, as they are supported by studies and results that demonstrate their effectiveness in improving functionality, such as mirror therapy (MT), cognitive therapeutic exercise (CTE), and task-oriented training. The aim of this study was to assess the effectiveness of MT and CTE combined with task-oriented training on the functionality, sensitivity, range, and pain of the affected upper limb in patients with acute stroke. A longitudinal multicenter study recruited a sample of 120 patients with acute stroke randomly and consecutively, meeting specific inclusion and exclusion criteria. They were randomly allocated into three groups: a control group only for task-oriented training (TOT) and two groups undergoing either MT or CTE, both combined with TOT. The overall functionality of the affected upper limb, specific functionality, sensitivity, range of motion, and pain were assessed using the Fugl–Meyer Assessment Upper Extremity (FMA-UE) scale validated for the Spanish population. An initial assessment was conducted before the intervention, a second assessment after completing the 20 sessions, and another three months later. ANCOVA analysis revealed statistically significant differences between the assessments and the experimental groups compared to the control group, indicating significant improvement in the overall functionality of the upper limb in these patients. However, no significant differences were observed between the two experimental groups. The conclusion drawn was that both therapeutic techniques are equally effective in treating functionality, sensitivity, range of motion, and pain in the upper limb following a stroke.

## 1. Introduction

Stroke, also known as cerebrovascular accident (CVA), is one of the most severe and common medical emergencies worldwide. It is of vascular origin, causing signs of neurologic deficit with rapid onset. These clinical signs can be focal or global, and if they last more than 24 h, without a clear cause that can cause death [1,2]. Since 1990, the incidence of strokes has increased by 70%, and deaths from strokes has increased by 43%, with a worrying rising trend in those under 70 years of age [3]. Today, this disease is the second leading cause of death worldwide and ranks third in terms of mortality and disability [2]. In 2019, there were 12.2 million strokes and 101 million prevalent strokes [4,5]. The vast majority of stroke cases occur due to potentially modifiable risk factors, which demonstrates the huge work that remains to be done to improve the prevention of this disease by reducing exposure to risk factors such as hypertension, diabetes, tobacco use, sedentary lifestyle, and abdominal fat [5,6]. In Spain, stroke is considered the leading cause of disability in adulthood and the second cause of dementia, significantly reducing the quality of life of patients and therefore that of their close social circle, directly affecting the health system [2].

One of the main sequelae resulting from a stroke is the loss of functionality, which can be significant, especially concerning the affected upper limbs. Following a stroke episode, it is common to experience a decline in function in the upper limbs, characterized by difficulties in performing activities of daily living (ADLs) or instrumental activities of daily living (IADLs) [7,8]. This loss of function may be related to muscle weakness, lack of motor coordination, or an inability to control movements. Motor impairment in the upper extremities occurs in approximately 80% of survivors, with 50% reporting pain in the upper limb during the first 12 months after the episode [9,10,11].

Furthermore, sensitivity in the affected limb can also be altered. Issues with tactile sensitivity may arise, such as decreased or loss of touch sensation, along with changes in temperature or pressure perception. These sensory changes can hinder precise movement execution or object recognition through the sense of touch. Similarly, the range of motion, i.e., the ability to move the joints of the upper limb, may also decrease after a stroke. This might manifest as restricted natural movements in the shoulder, elbow, wrist, or fingers, further complicating the performance of everyday tasks [12,13].

Pain affecting the upper limbs is another common outcome after a stroke. Pain can be neuropathic or related to posture, movement, and muscle stiffness. The presence of pain can negatively impact rehabilitation and the ability to perform exercises or therapies aimed at recovering limb functionality [12,14].

All of these factors together can trigger hemiplegia or hemiparesis, common conditions after a stroke involving paralysis or muscle weakness on one side of the body, primarily affecting an upper and lower limb on the same side. This signifies a change in the ability to achieve a normal level of muscle strength, including sensory alteration, loss of motor control, and spasticity [15,16]. This condition can significantly impact a person’s functionality. Hemiplegia can range from mild weakness to complete paralysis on one side of the body, affecting the ability to move and perform daily tasks.

Collectively, these effects can complicate daily life and the recovery process after a stroke. Rehabilitation in these cases usually aims to address these challenges, seeking to improve function, sensitivity, and range of motion and to manage pain to regain the maximum possible functionality of the affected upper limb. This post-stroke rehabilitation period should commence as early as possible, with function recovery predominantly occurring in the first few weeks [17], although there are studies indicating that patients go through a phase known as spontaneous recovery during the initial weeks [18].

Before initiating rehabilitation, conducting a thorough assessment to determine the stroke’s aftermath is crucial. Understanding where to begin is vital for implementing quality rehabilitation. To achieve this, there are several instruments used to assess the functional state of stroke survivors. The Fugl–Meyer Assessment—Upper Extremity (FMA-UE) scale is currently the most widely used quantitative evaluation to measure functionality and motor recovery post-stroke [12,19,20].

Regarding rehabilitation, therapies supported by scientific evidence play a crucial role in this process, as they are backed by studies and outcomes demonstrating their effectiveness in improving functionality and recovering lost skills. Some of these include mirror therapy (MT), cognitive therapeutic exercise (CTE), and task-oriented training.

MT is a rehabilitation technique that utilizes visual illusion to enhance motor function in individuals who have experienced strokes, limb injuries, or lost functionality in a limb. This therapy involves using a mirror to create the illusion that the affected limb is functioning normally [21]. A mirror is positioned to reflect the unaffected limb, while the affected limb remains hidden behind it. Moving the unaffected limb creates a reflection in the mirror, simulating movement in the affected limb, tricking the brain into perceiving normal movement. This therapy focuses on repeating controlled and specific movements, which may promote neuroplasticity—the brain’s ability to reorganize and adapt through experience and repetitive practice. It is believed that this therapy could help restore motor function, enhance coordination, and alleviate chronic pain associated with the affected limb [11,22,23].

As for CTE, also known as the Perfetti method, is a neurorehabilitation approach offering personalized and specific treatment for each patient. Its goal is to recover lost or altered movement due to central nervous system damage. This method involves assigning the patient a specific problem-solving task that can be resolved through fragmented movement of body segments guided by the therapist. CTE aims to improve the specific motor deficit in the hemiplegic upper limb by addressing patterns such as abnormal reactions to stretching, abnormal irradiation, motor mobility of elementary schemes, and promoting efficient and high-quality motor recruitment. Essentially, it aims to reactivate and strengthen neural connections damaged by stroke [24,25].

On the other hand, these patients can benefit from task-oriented training, which is an effective way to encourage and develop motor skills and brain plasticity through the repetition of specific and functional tasks. It relies on tailored and personalized activities that mimic daily actions. Therapists design specific training programs for each patient, considering their individual needs, motor deficiencies, and recovery goals. These programs focus on activities resembling the tasks the patient needs to perform in their daily life. The effectiveness of task-oriented training lies in its emphasis on functionality and practical application of motor skills in real-life situations. This approach aims not only to restore motor function but also to improve the patient’s independence in daily activities, which can have a significant impact on their quality of life [26].

Recent studies have demonstrated that the combined use of these therapies activates central nervous system plasticity more effectively than when used individually, to improve motor function [27,28,29,30]. However, to date and to the authors’ knowledge, there is no article comparing if any of these combinations (MT or CTE combined with task-oriented training) are the most effective in improving upper limb function after a stroke.

Therefore, the aim of the present study is to verify the effectiveness of combining these techniques on upper limb functionality after a stroke and to determine which of them yields better results.

## 2. Materials and Methods

### 2.1. Study Design and Participants

This longitudinal, multicenter study was conducted in collaboration with the University of Burgos (UBU), Burgos University Hospital (HUBU), San Juan de Dios Hospital (Burgos), and Reina Sofía Hospital in Córdoba (Spain).

The inclusion criteria applied are as follows in Table 1.

### 2.2. Procedure

The sample for this study was recruited upon discharge from the stroke units and neurology services of the hospitals using consecutive sampling.

The study design was a randomized, controlled, and singled blinded clinical trial with three groups: control group (CG), experimental 1 (EG1), and experimental 2 (EG2). The participants were recruited by the rehabilitation doctor, who assessed whether or not they met the inclusion criteria for the study. Participants were randomly assigned to groups in a 1:1:1 ratio using a masking process, centrally generated by an independent investigator using Epidat 4.2 (freely available software for epidemiological analysis, which, among other things, allows random assignation) before participant inclusion. Likewise, different researchers conducted the administration of therapies to the experimental groups. The professionals responsible for applying the therapies in the different centers were occupational therapists and physiotherapists specialized and trained in the techniques used. Participants underwent an initial evaluation one month after the stroke, where inclusion and exclusion criteria were applied. Subsequently, after collaboration agreements with participating centers, data collection commenced. The patients, at the time of recruitment, were outpatients who attended the hospital for rehabilitation prescribed by their doctor.

During the intervention, conducted over 20 sessions, 5 days a week [31,32,33], groups were divided so that participants in group EG1 received MT combined with task-oriented training, and participants in group EG2 received CTE combined with task-oriented training. The therapies were divided, dedicating 30 min to either MT or CTE, and the remaining 30 min to task-oriented training. Thus, all three groups followed their usual therapy prescribed by the rehabilitation physician (task-oriented training), and the EG received 20 additional treatment sessions in which MT and CTE were applied. The therapy was carried out in the aforementioned hospitals by either an occupational therapist or a physiotherapist.

A second post-intervention assessment and a follow-up visit three months after the second evaluation were conducted to assess patient progress during the subacute phase of recovery. In the following Figure 1, a flowchart is depicted summarizing the procedure.

The research plan was approved by the IR Approval Committee of HUBU 2134/2019. Data collection was carried out in participating centers by designated personnel, and the data were anonymized before sharing with the research team, who remained anonymous and aggregated from that point forward.

### 2.3. Instruments

The independent variables collected encompassed sociodemographic and clinical data, such as age, gender, number of children, or place of residence. Additionally, specific clinical variables following the stroke were considered, such as the affected and dominant side, and upper limb functionality.

To assess functional capacity, the FMA-UE [34,35,36] was utilized. This scale, validated in the Spanish population and translated into Spanish, demonstrates high reliability and validity, with a Cronbach’s alpha of 0.973. It consists of 33 items divided into 3 domains (motor, sensory, and range of motion and pain). Each item is scored from 0 (not performed) to 2 (complete execution), with a total score of 66 points, of which 36 are assigned to the proximal part of the arm and 30 to the wrist and hand, 12 points for the sensory domain and 48 for the range of motion and pain domain; thus, resulting in a total scale score of 126 points. Higher scores on this scale reflect greater functionality in the upper limbs, as well as normal exteroceptive and proprioceptive sensitivity, an adequate range of passive mobility, and the absence of pain. Page, Fulk, and Boyne [37] established that the clinically important difference of FMA-UE scored ranges from 4.25 to 7.25 points depending on the different factors of the scale, while Hiragami and Harada [34] concluded that a score of 12.4 is the minimal clinically important difference and that it is likely to be perceived as significant by stroke patients with moderate to severe hemiparesis.

### 2.4. Intervention

#### 2.4.1. Mirror Therapy

In MT treatment, the patient sat in a chair with forearms resting on a table, while a mirror was positioned between both arms at a right angle to the torso. The affected limb was placed behind the mirror, out of the patient’s visual field, in a comfortable position. The healthy limb was positioned similarly to be reflected without distortion in the mirror, removing any visible object or symbol solely for the healthy limb (Figure 2).

MT was employed in three different modes. In the first mode, the patient attempted to mirror the movement of the healthy hand with the affected hand in synchronized fashion. In the second mode, the patient imagined the reflected movement of the healthy hand being performed by the affected hand. In the third mode, the therapist assisted the patient’s affected hand to replicate the movement of the healthy hand. Exercises began with simple movements without objects in the initial sessions, progressed to movements with objects in the intermediate sessions, and finally included more complex movements with objects in the later sessions. The progression was tailored individually to the patient’s recovery, starting with initial imagination and moving towards execution assisted by the therapist. All exercises were performed slowly and repeated at least 15 times, adjusting complexity according to each patient’s individual capabilities and limitations. Likewise, all participants applied this intervention methodology through the three modes described above.

#### 2.4.2. Cognitive Therapeutic Exercise

CTE are divided into three levels: first, second, and third degree. Initially, all patients performed first-degree exercises until they achieved control over the stretch reflex, regulating intensity, duration, and location. Once this was mastered, they moved on to second-degree exercises. Subsequently, third-degree exercises were implemented, where the patient learned to adjust movements based on perceptual hypotheses, after automating control over abnormal second-degree motor behaviors.

At the first level, excessive stretch reflex reactions (spasticity) and reduced sensitivity were addressed, with the therapist performing movements alongside the patient. These exercises required the patient’s active attention at all times. In the second level, the goal was to control involuntary activation of muscle groups (abnormal irradiation). Here, the patient performed movements with minimal therapist assistance, using different tactile, kinetic, weight, grip, and friction stimuli combined with first-degree exercises. At the third level, the focus was on voluntary movement control, its fragmentation, variability, and adaptation, aiming for complete automation of movements without any therapeutic assistance from the therapist.

#### 2.4.3. Task-Oriented Training

Task-oriented training was structured sequentially, adapting in each session to replicate real-life situations. Complex activities were broken down into simpler tasks to facilitate learning. Activities included meal preparation and organization, handling upper and lower garments, and personal hygiene tasks like brushing teeth, combing hair, shaving, or applying makeup. Short rest periods were interspersed, and task difficulty was progressively increased to enhance performance.

### 2.5. Statistical Analysis

Descriptive analyses were performed on the sample characteristics, expressing categorical variables in absolute frequencies and percentages, and continuous variables in means and standard deviations (SD). The normality of the dataset was assessed using the Kolmogorov–Smirnov test. To assess differences between groups across various assessments conducted during the intervention, an ANCOVA analysis was employed. The treatment group served as the fixed factor, while the differential scores of the evaluated variables—such as overall upper limb functionality measured with FMA-UE and the subscales assessing specific functionality, sensitivity, range, and upper limb pain—were utilized as dependent variables. The pre-test scores of the same variables were used as covariates.

To determine the sample size, a formula adjusted for finite populations was employed, considering a known proportion of stroke cases in the population based on data from the National Institute of Statistics (INE) [38], with an estimated 1% margin of error. The conclusion was that the sample should consist of 81 stroke patients.

Statistical analysis was conducted using SPSS version 25 (IBM Inc., Chicago, IL, USA). A significance level of *p* < 0.05 was set for all statistical analyses.

## 3. Results

Data from a total of 120 patients, one month after suffering a stroke, were analyzed. The mean age was 68.92 (SD = 11.79), with an age range from 41 to 96 years. The gender distribution was 58.3% for males (*n* = 70) and 41.7% for females (*n* = 50).

The majority of the sample was collected in Burgos (74.8%), and most participants were right-handed (95.8%).

Finally, the sample distribution across treatment groups was entirely equitable, with 40 patients in each group, consisting of the control group, MT group, and CTE group.

Of these, 51.7% (*n* = 62) had the left upper limb affected, while 48.3% (*n* = 58) had their right upper limb affected. Additionally, 92.5% of them experienced an ischemic stroke, with 6.7% (*n* = 8) being hemorrhagic strokes.

In Table 2, significant or highly significant differences can be observed between the differential scores of the control group with the two experimental groups in all measured variables, except for range of motion and pain domain, at the second evaluation. However, no statistically significant differences are observed between the two experimental groups (EG), except for range of motion and pain domain, where there are significant differences.

In Table 3, significant or highly significant differences were observed between the differential scores of the control group with the two experimental groups in all the variables, except for range of motion and pain domain. However, statistically significant differences between both experimental groups were not observed in any case.

In the following Figure 3, the raw results obtained by the CG, CTE, and MT in the three evaluations for functionality, the motor domain, sensory domain, and range of motion and pain domain are representatively displayed.

Finally, no differences were observed between the differential scores of the control group with the two experimental groups or between the two experimental groups when comparing second and third evaluations.

## 4. Discussion

The aim of this study was to assess the effectiveness of MT and CTE combined with task-oriented training on the functionality of the affected upper limb in acute stroke patients. Our results indicate a significant improvement between the first and second evaluation and between the first and third evaluation after applying these combined therapies with task-oriented training over a 20-session protocol compared to our CG, which only received their usual therapy. The combination of task-oriented training with mirror therapy or cognitive therapeutic exercise obtained similar results, better than in the usual treatment control group. The improvement seems to be sustained over time.

The statistically significant differences in global limb functionality, motor domain and sensory domain are consistent, with several studies indicating that intervention using MT and CTE improves the total FMA-EU score, thus increasing upper limb functionality and motor domain, which in turn improves motor function and ADLs [39,40].

Furthermore, significant and highly significant differences were observed between the CG and the two EG, indicating a superior improvement for both EG compared to the CG, except for the range of motion and pain domain. These results are consistent with other studies suggesting that groups incorporating MT and cognitive exercises show greater motor and functional recovery and better self-care compared to CG [40,41,42].

Our results also demonstrated that in the long term, between the first and third evaluations, approximately 5 months after the initial measurement and after the intervention had ended, there were still statistically significant differences in the measured variables. A significant improvement over time was observed, suggesting a sustained improvement even after the intervention had concluded. Specifically, highly significant differences were observed in the overall functionality of the affected upper limb and in the motor domain of the FMA-UE scale in stroke patients. Additionally, significant differences were observed in the sensory domain, but no differences were found in the range of motion and pain domain.

On one hand, this study demonstrates that MT training brings about improvements in motor performance in stroke patients with upper limb motor dysfunction. The improvements reported in this research on motor and functional recovery through MT use are also supported by findings described in similar research [41,42,43,44]. Additionally, changes in neuroplasticity are crucial for motor function recovery [45], acknowledging that task-oriented training is an effective way to reduce disability rates by fostering functional brain reorganization. However, many times when patients receive traditional treatment, they lack attention and initiative, negatively impacting the activation of the corresponding brain cortex, thereby affecting neuroplasticity and functional reorganization [46]. In this regard, MT necessitates active movement of bilateral upper limbs, which can enhance patient initiative [40]. Furthermore, it also requires simultaneously observing the reflection in the mirror of normal limb movements, potentially enhancing patient focus, favoring more cortical activation than traditional rehabilitation training.

Regarding CTE, the findings of this current research supporting improved upper limb functional capacity in acute stroke patients are consistent with results presented in previous studies within the same field [25,46]. These same authors have also shown that CTE can enhance stretching speed, rapid reaction, distance, as well as stimulate cognitive processes to perceive precise movements and halt abnormal elements to promote appropriate movements. Similarly, Morioka et al. [47] discovered that during spatial tasks, bursts of airflow significantly increased activation in the premotor cortex for motor learning and in the dorsolateral prefrontal cortex for occupational memory function. This aspect plays a crucial role in adjusting movement and muscle activity as well as in learning and remembering exercise methods [48]. Hence, recovery from abnormal motor functions might be more effective with cognitive task training that uses contact and spatial tasks. CTE differs from other treatment strategies in the internal observation of changes in cognitive processes. Both qualitative and quantitative elements are used to determine aspects related to movement, language, attention, imagery, and learning. Furthermore, the recovery of motor function is associated with the brain’s cognitive processes, and the quality of recovery depends on whether these cognitive components have been exercised [49].

Other studies also indicate that improvements in functionality were sustained up to 6 months after the intervention ended [50]. There are even studies suggesting that improvements in upper limb functionality can be maintained for a period of 3 months to 1 year following the conclusion of the intervention [51].

However, in our study, no statistically significant differences were observed between both experimental groups that received MT and CTE combined with task-oriented training. This is not in line with other studies, which indicate that MT yields better results in performing ADLs, a greater reduction in pain, and enhanced motor function compared to therapies like CTE. Moreover, patients receiving MT reportedly have higher FMA scores compared to those receiving other types of therapy [11].

Some research states that it is not solely about the type of intervention but also about the duration of the therapy on which the results obtained may depend. Studies suggest that with a higher dosage of intervention, there are greater functional improvements [52]. In our research, both EGs received higher dosages of intervention, so that is also a variable to take into account.

The study faces several limitations, such as the sample size, making it challenging to generalize the results. Perhaps modifying the inclusion and exclusion criteria could facilitate sample acquisition, as the recruitment of patients meeting all proposed criteria was quite challenging. This research did not include participants exhibiting hemineglect, Wernicke’s aphasia, or mixed aphasia or visual impairment (homonymous hemianopsia) so the results of this study would not be applicable to them. In addition, the total dose of treatment received was higher in the experimental groups than in the control group because the therapies under study were implemented at higher doses than the therapies the patients had already been prescribed, so it is necessary to take the results into consideration within this fact. Finally, particular attention should be paid to the fact that the study was initiated prior to the pandemic, which hindered patient recruitment for an extended period. As a result, many recruited patients experienced time before COVID, and several interrupted their treatments due to infection, which may have interfered with the results.

As strong points of the study, it is worth highlighting a solid study design. A well-structured design with randomized allocation methods has been implemented, enhancing the study’s validity. Additionally, multiple assessments have been conducted, allowing for a comprehensive understanding of the rehabilitation impact. Long-term follow-ups have been performed to assess sustained effects of rehabilitation. Similarly, the sample has been gathered from various centers, broadening the representativeness, validity, and generalizability of the findings. As for future lines of research, long-term studies investigating the use of these same techniques for the treatment of chronic stroke patients are proposed, along with comparisons with other therapeutic methods. Assessing the long-term benefits and efficacy of these approaches is crucial. Additionally, exploring new technologies, such as virtual reality or neuroimaging, could complement and allow the observation of improvements in the effectiveness of these techniques, providing a better understanding of their impact on patient functionality.

## 5. Conclusions

In conclusion, our findings have suggested the effectiveness of MT and CTE techniques combined with task-oriented training in improving the overall functionality of the upper limb, as well as its specific functionality and sensitivity, but not for range of motion and pain.

We observed a significant improvement in all these variables following the treatment application, and these improvements remained over time. However, there were no significant differences observed between both treatment techniques concerning the improvement of the measured variables in the study. Therefore, it can be concluded that both techniques are equally effective for treating and rehabilitating the functionality, sensitivity, range of motion, and pain in the upper limb in stroke patients.

## Figures and Tables

**Figure 1 healthcare-12-00569-f001:**
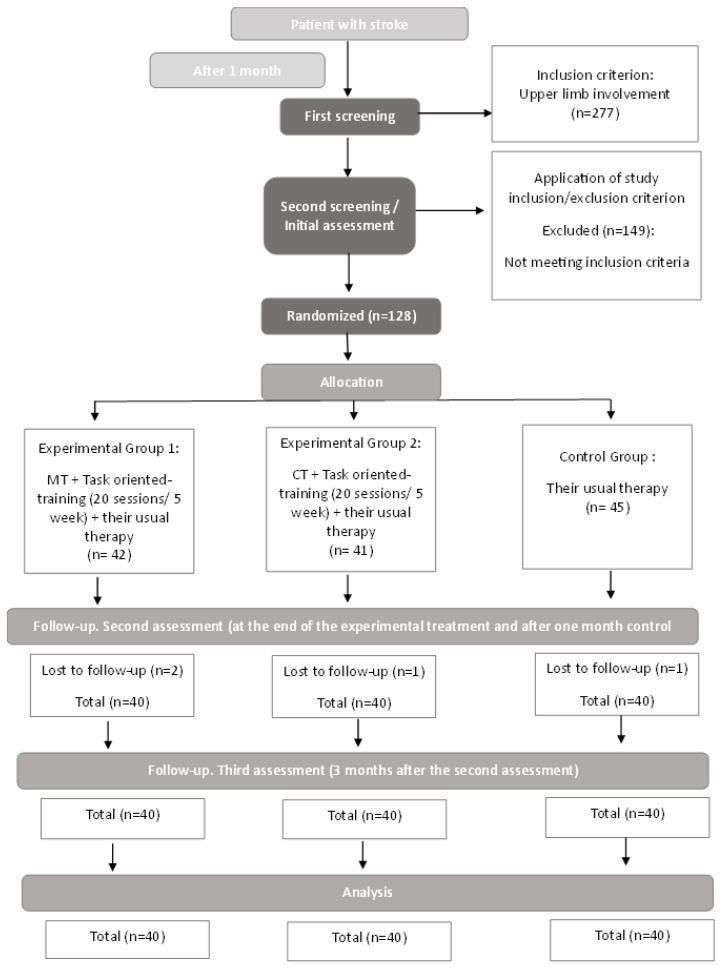
Flowchart.

**Figure 2 healthcare-12-00569-f002:**
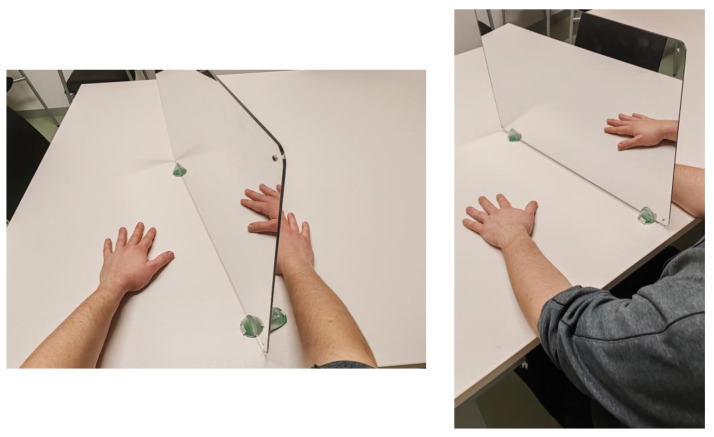
Example of patient positioning for MT.

**Figure 3 healthcare-12-00569-f003:**
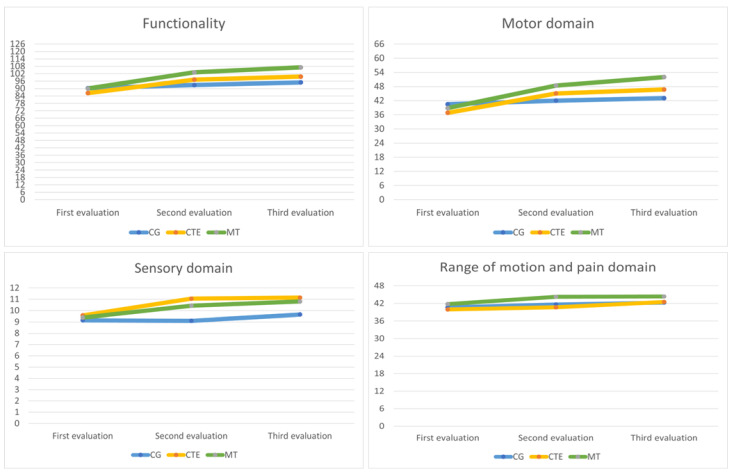
Raw scores for the CG, CTE, MT groups in the three evaluations.

**Table 1 healthcare-12-00569-t001:** Inclusion and exclusion criteria.

**Inclusion Criteria**	
1	Being over 18 years old
2	Having been diagnosed with residual hemiparesis due to an ischemic or hemorrhagic stroke
3	Presenting a level of movement in the affected upper limbs within stages II to IV according to the Brunnstrom scale [31]
4	Obtaining a score in the Montreal Cognitive Assessment (MoCA) equal to or higher than 26 [32,33]
5	Obtaining informed consent from all participants
**Exclusion criteria (based on diagnostic information provided by the neurologist’s clinical evaluation)**	
1	Participants exhibiting hemineglect
2	Wernicke’s aphasia or mixed aphasia
3	Visual impairment (homonymous hemianopsia)

**Table 2 healthcare-12-00569-t002:** ANCOVA analysis between treatment group and first-second evaluation.

Variables	Group	First Evaluation Mean (SD)	Second Evaluation Mean (SD)	Treatment Group	Mean Difference	SD	*p*	95% CI		Observed Power
								LI	LS	
Functionality	CG	90.25 (29.978)	92.78 (29.873)	CTE	−7.90	2.264	0.001 **	−12.392	−3.425	0.994
				MT	−10.56	2.260	<0.001 **	−15.041	−6.088	
	CTE	86.33 (26.774)	97.23 (24.433)	CG	7.90	2.264	0.001 **	3.425	12.392	
				MT	−2.65	2.263	0.243	−7.139	1.827	
	MT	89.95 (27.964)	103.08 (25.998)	CG	10.56	2.260	<0.001 **	6.088	15.041	
				CTE	2.65	2.263	0.243	−1.827	7.139	
Motor domain	CG	40.48 (21.903)	42.03 (22.208)	CTE	−6.28	1.919	0.001 **	−10.086	−2.485	0.977
				MT	−7.88	1.915	<0.001 **	−11.677	−4.092	
	CTE	36.85 (20.314)	45.05 (18.474)	CG	6.28	1.919	0.001 **	2.485	10.086	
				MT	−1.59	1.915	0.406	−5.392	2.195	
	MT	38.83 (19.945)	48.43 (20.671)	CG	7.88	1.915	<0.001 **	4.092	11.677	
				CTE	1.59	1.915	0.406	−2.195	5.392	
Sensory domain	CG	9.15 (3.585)	9.10 (3.720)	CTE	−1.67	0.425	<0.001 **	−2.512	−0.827	0.955
				MT	−1.17	0.425	0.007 *	−2.018	−0.335	
	CTE	9.57 (3.720)	11.05 (2.062)	CG	1.67	0.425	<0.001 **	0.827	2.512	
				MT	0.49	0.425	0.248	−0.348	1.335	
	MT	9.38 (3.600)	10.43 (3.145)	CG	1.17	0.425	0.007 *	0.335	2.018	
				CTE	−0.49	0.425	0.248	−1.335	0.348	
Range of motion and pain domain	CG	40.63 (10.883)	41.65 (9.919)	CTE	0.46	0.903	0.612	−1.328	2.247	0.595
				MT	−1.69	0.903	0.063	−3.486	0.092	
	CTE	39.90 (8.924)	40.63 (9.903)	CG	−0.46	0.903	0.612	−2.247	1.328	
				MT	−2.16	0.905	0.019 *	−3.949	−0.364	
	MT	41.75 (9.009)	44.23 (6.379)	CG	1.69	0.903	0.063	−0.092	3.486	
				CTE	2.16	0.905	0.019 *	0.364	3.949	

CG: control group; MT: mirror therapy; CTE: cognitive therapeutic exercise; SD: standard deviation. * *p* < 0.05; ** *p* < 0.001.

**Table 3 healthcare-12-00569-t003:** ANCOVA analysis between treatment group and first-third evaluation.

Variables	Group	First Evaluation Mean (SD)	Third Evaluation Mean (SD)	Treatment Group	Mean Difference	SD	*p*	95% CI		Observed Power
								LI	LS	
Functionality	CG	90.25 (29.978)	94.95 (27.556)	CTE	−7.78	2.815	0.007 *	−13.356	−2.203	0.983
				MT	−12.41	2.811	<0.001 **	−17.978	−6.843	
	CTE	86.33 (26.774)	99.65 (25.172)	CG	7.78	2.815	0.007 *	2.203	13.356	
				MT	−4.63	2.815	0.103	−10.206	0.944	
	MT	89.95 (27.964)	107.13 (23.516)	CG	12.41	2.811	<0.001 **	6.843	17.978	
				CTE	4.63	2.815	0.103	−0.944	10.206	
Motor domain	CG	40.48 (21.903)	43.05 (20.539)	CTE	−6.63	2.165	0.003 *	−10.920	−2.343	0.993
				MT	−10.28	2.161	<0.001 **	−14.564	−6.004	
	CTE	36.85 (20.314)	46.75 (18.648)	CG	6.63	2.165	0.003 *	2.343	10.920	
				MT	−3.65	2.161	0.094	−7.934	0.628	
	MT	38.83 (19.945)	52.00 (18.753)	CG	10.28	2.161	<0.001 **	6.004	14.564	
				CTE	3.65	2.161	0.094	−0.628	7.934	
Sensory domain	CG	9.15 (3.585)	9.65 (3.585)	CTE	−1.26	0.468	0.008 *	−2.192	−0.339	0.721
				MT	−1.02	0.467	0.030 *	−1.952	−0.100	
	CTE	9.57 (3.720)	11.15 (2.155)	CG	1.26	0.468	0.008 *	0.339	2.192	
				MT	0.24	0.467	0.609	−0.686	1.165	
	MT	9.38 (3.600)	10.80 (2.747)	CG	1.02	0.467	0.030 *	0.100	1.952	
				CTE	−0.24	0.467	0.609	−1.165	0.686	
Range of motion and pain domain	CG	40.63 (10.883)	42.25 (9.647)	CTE	−0.711	1.230	0.564	−3.147	1.724	0.150
				MT	−1.359	1.231	0.272	−3.797	1.079	
	CTE	39.90 (8.924)	42.50 (7.643)	CG	0.711	1.230	0.564	−1.724	3.147	
				MT	−0.647	1.233	0.601	−3.090	1.795	
	MT	41.75 (9.009)	44.33 (7.180)	CG	1.359	1.231	0.272	−1.079	3.797	
				CTE	0.647	1.233	0.601	−1.795	3.090	

CG: control group; MT: mirror therapy; CTE: cognitive therapeutic exercise; SD: standard deviation. * *p* < 0.05; ** *p* < 0.001.

## Data Availability

The data that support the findings of this study are available from the corresponding author, S.A.-P., upon reasonable request.

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
