# Peer review of "Efficacy of a Rehabilitation Program Using Mirror Therapy and Cognitive Therapeutic Exercise on Upper Limb Functionality in Patients with Acute Stroke"

_healthcare, 2024, doi:10.3390/healthcare12050569_

Round 1

Reviewer 1 Report

Comments and Suggestions for Authors

good study but I would make several changes:

- In the abstract I would not start directly with the objective, I would explain the topic and then the stated objective. line 14.

- In the introduction I would put the objective in a separate paragraph.

- In the methodology, the inclusion and exclusion criteria would be put with tables that are more explicit and visual. lines 137-145

- in the explanations of the interventions I would tmabein make it more visual and not with so many letters.

- below the limitations in the discussion I would add the strengths of the study

Congratulations on the work

Author Response

Mr. Jessica Fernández Solana
Department of health sciences
University of Burgos, Paseo Comendadores s/n.
Burgos, 09001, Spain
Tel. (+34) 947499108
Email: jfsolana@ubu.es
12-01-2024
Healthcare. Subject: Submissions Needing Revision
Dear editor.
Thank you very much for inviting us to submit our response to reviewers for our manuscript (healthcare-2797806) entitled: “Efficacy of a Rehabilitation Program Using Mirror Therapy and Cognitive Therapeutic Exercise on Upper Limb Functionality in Patients with Acute Stroke”
We have checked our manuscript according to the Academic Editor, the reviewers’ comments and the Journal requirements. We have also responded to some comments from reviewers point by point).
We would be very grateful if you could consider our manuscript to be published in your journal.
Yours sincerely,
Jessica Fernández Solana, OT, PT
A. Response to Reviewer 1:
First of all, we would like to express our sincere gratitude for all comments and suggestions received from the Reviewer 1. This information has certainly enriched the text for its best understanding, thank you very much indeed. We have clarified the reviewer1’s questions. We have introduced the required changes both in our answers to the specific comments and in the final manuscript V2.
good study but I would make several changes:
- In the abstract I would not start directly with the objective, I would explain the topic and then the stated objective. line 14.
Response: thank you for your comment, a paragraph has been added at the beginning of the abstract.
- In the introduction I would put the objective in a separate paragraph.
Response: thank you for your comment, the objective has been put in a separate paragraph.
- In the methodology, the inclusion and exclusion criteria would be put with tables that are more explicit and visual. lines 137-145
Response: we have written these criteria in a table, it now seems more explicit and visual as you stated.
- in the explanations of the interventions I would make it more visual and not with so many letters.
Response: thank you for your comment, we have tried to modify the wording to make it more explanatory.
- below the limitations in the discussion I would add the strengths of the study
Response: thank you for your comment, some of the strengths of the study have been added (see lines 418-424).
Congratulations on the work
We hope we have now answered all your comments and we are looking forward to hearing from you again.
Thank you very much,
Jessica Fernández Solana, OT, PT

Reviewer 2 Report

Comments and Suggestions for Authors

I congratulate you for your effort and your sample size. It is an original investigation on a important topic.

I consider it appropriate to make modifications to the Abstract, Discussion and Conclusions, for the following reasons:

- In ABSTRACT, it is said: "The conclusion reached was that both therapeutic techniques are equally effective to treat functionality, sensitivity, range of motion and pain in the upper limb after a stroke." Are both therapeutic techniques only MT and CTE?

It could be stated whether there was another fourth group with only task-oriented training treatment, with lower results. But with the groups studied it can only be stated: "the combination of one of the two therapeutic techniques, with task-oriented training, showed similar positive results."

- And in DISCUSSION it should be commented that the good results obtained in both groups, similar, could be due exclusively to task-oriented training - because there is no fourth group of only task-oriented training with which compare them-.

And reflect it in the Conclusions, since it has not been evaluated whether the improvement is due to task-oriented training or to the other therapy that was applied to each of the experimental groups.

Or say: "The combination of task-oriented training with mirror therapy or cognitive therapeutic exercise obtained similar results, better than in the usual treatment control group".

- In the Control Group: could you specify what the usual treatment was like? Did they not receive task-oriented training?

- It would be advisable to expand the Discussion that another limitation of the study is that it has not included these pathologies from the exclusion criteria: "(1) participants exhibiting hemineglect, (2) Wernicke's aphasia or mixed aphasia, and (3) visual impairment (homonymous hemianopsia)".

- I consider that in Results and Discussion reference should be made to the fact that:

In TABLE 2, there are no significant differences between CG and the two experimental groups in "Range of motion and pain domain" (p=0.612 and p=0.063). But there is a statistically significant difference between CTE and MT (p=0.019).

And in TABLE 3 there are also no statistically significant differences between CG and the two experimental groups in "Range of motion and pain domain". Therefore I do not understand why it says in line 307-8 of the DISCUSSION: "...and significant enhancement in the range of motion and pain in the affected limb."

In papers, the term "significant" should only be used when it is statistically significant.

Author Response

Mr. Jessica Fernández Solana
Department of health sciences
University of Burgos, Paseo Comendadores s/n.
Burgos, 09001, Spain
Tel. (+34) 947499108
Email: jfsolana@ubu.es
12-01-2024
Healthcare. Subject: Submissions Needing Revision
Dear editor.
Thank you very much for inviting us to submit our response to reviewers for our manuscript (healthcare-2797806) entitled: “Efficacy of a Rehabilitation Program Using Mirror Therapy and Cognitive Therapeutic Exercise on Upper Limb Functionality in Patients with Acute Stroke”
We have checked our manuscript according to the Academic Editor, the reviewers’ comments and the Journal requirements. We have also responded to some comments from reviewers point by point).
We would be very grateful if you could consider our manuscript to be published in your journal.
Yours sincerely,
Jessica Fernández Solana, OT, PT
B. Response to Reviewer 2:
First of all, we would like to express our sincere gratitude for all comments and suggestions received from the Reviewer 2. This information has certainly enriched the text for its best understanding, thank you very much indeed. We have clarified the reviewer2’s questions. We have introduced the required changes both in our answers to the specific comments and in the final manuscript V2.
I congratulate you for your effort and your sample size. It is an original investigation on a important topic.
I consider it appropriate to make modifications to the Abstract, Discussion and Conclusions, for the following reasons:
- In ABSTRACT, it is said: "The conclusion reached was that both therapeutic techniques are equally effective to treat functionality, sensitivity, range of motion and pain in the upper limb after a stroke." Are both therapeutic techniques only MT and CTE?
Response: thank you for your comment. In material and methods we have made modifications to clarify this question for the whole manuscript. Both intervention groups receive task-oriented therapy equally. What differentiates the two intervention groups are MT and CTE.
It could be stated whether there was another fourth group with only task-oriented training treatment, with lower results. But with the groups studied it can only be stated: "the combination of one of the two therapeutic techniques, with task-oriented training, showed similar positive results."
Response: thank you for your comment. There was no fourth group with task-oriented training. There were only two intervention groups, one with MT and one with CTE, and in both of them task-oriented therapy was applied in a complementary way.
The effectiveness of each of these therapies separately has already been studied. What this study aims to demonstrate is the effectiveness of these therapies in combination.
- And in DISCUSSION it should be commented that the good results obtained in both groups, similar, could be due exclusively to task-oriented training - because there is no fourth group of only task-oriented training with which compare them-.
And reflect it in the Conclusions, since it has not been evaluated whether the improvement is due to task-oriented training or to the other therapy that was applied to each of the experimental groups.
Or say: "The combination of task-oriented training with mirror therapy or cognitive therapeutic exercise obtained similar results, better than in the usual treatment control group".
Response: We have added to the discussion the sentence that you propose (lines 324-326)
- In the Control Group: could you specify what the usual treatment was like? Did they not receive task-oriented training?
Response: the control group only was evaluated. Patients in the control group were assessed at three different times. This information has been indicated in the material and methods section. These patients did not receive therapy from us, but they did receive their usual therapy prescribed by their rehabilitation physician.
- It would be advisable to expand the Discussion that another limitation of the study is that it has not included these pathologies from the exclusion criteria: "(1) participants exhibiting hemineglect, (2) Wernicke's aphasia or mixed aphasia, and (3) visual impairment (homonymous hemianopsia)".
Response: Thank you for this comment, we also think it is important to reflect this; we have added “This research did not include participants exhibiting hemineglect, Wernicke's aphasia or mixed aphasia or visual impairment (homonymous hemianopsia) so the results of this study would not be applicable to them.” In the limitations paragraph.
- I consider that in Results and Discussion reference should be made to the fact that:
In TABLE 2, there are no significant differences between CG and the two experimental groups in "Range of motion and pain domain" (p=0.612 and p=0.063). But there is a statistically significant difference between CTE and MT (p=0.019).
And in TABLE 3 there are also no statistically significant differences between CG and the two experimental groups in "Range of motion and pain domain". Therefore, I do not
understand why it says in line 307-8 of the DISCUSSION: "...and significant enhancement in the range of motion and pain in the affected limb."
In papers, the term "significant" should only be used when it is statistically significant.
Response: We have modified this in the text and/or tables, both in results and the discussion.
We hope we have now answered all your comments and we are looking forward to hearing from you again.
Thank you very much,
Jessica Fernández Solana, OT, PT

Reviewer 3 Report

Comments and Suggestions for Authors

Thank you for the opportunity to review your manuscript. The topic is relevant and of paramount importance. After reviewing your manuscript, I would like to make some suggestions that I believe will improve the reporting of your work. These suggestions refer exclusively to the report and not to the merit of the research.

Ln 140: (4) (4) is repeated

Ln 154: please explain blinding

Ln 160: "EG1 received TE..." do you mean MT?

ln 179: number of children and place of residence. How are these data relevant for the analysis?

Results: please always acknowledge "n" for each percentage; do not repeat in the tables data already presented in the text.

Ln 394: due to study limitations, isn´t it more accurate to use the term "suggests" instead of "demonstrated"?

Ln 418: data availability refers to where the research data associated with the paper is available and not to informed consent.

Author Response

Mr. Jessica Fernández Solana
Department of health sciences
University of Burgos, Paseo Comendadores s/n.
Burgos, 09001, Spain
Tel. (+34) 947499108
Email: jfsolana@ubu.es
12-01-2024
Healthcare. Subject: Submissions Needing Revision
Dear editor.
Thank you very much for inviting us to submit our response to reviewers for our manuscript (healthcare-2797806) entitled: “Efficacy of a Rehabilitation Program Using Mirror Therapy and Cognitive Therapeutic Exercise on Upper Limb Functionality in Patients with Acute Stroke”
We have checked our manuscript according to the Academic Editor, the reviewers’ comments and the Journal requirements. We have also responded to some comments from reviewers point by point).
We would be very grateful if you could consider our manuscript to be published in your journal.
Yours sincerely,
Jessica Fernández Solana, OT, PT
C. Response to Reviewer 3:
First of all, we would like to express our sincere gratitude for all comments and suggestions received from the Reviewer 3. This information has certainly enriched the text for its best understanding, thank you very much indeed. We have clarified the reviewer3’s questions. We have introduced the required changes both in our answers to the specific comments and in the final manuscript V2.
Thank you for the opportunity to review your manuscript. The topic is relevant and of paramount importance. After reviewing your manuscript, I would like to make some suggestions that I believe will improve the reporting of your work. These suggestions refer exclusively to the report and not to the merit of the research.
Ln 140: (4) (4) is repeated
Response: thank you, I have removed the duplicate number.
Ln 154: please explain blinding
Response: thanks for your comment, the following information has been added on lines 159-167:
“The participants were recruited by the rehabilitation doctor, who assessed whether they met the inclusion criteria for the study. Participants were randomly assigned to groups using a masking process in a 1:1:1 ratio, centrally generated by an independent investigator using Epidat 4.2 before participant inclusion. Likewise, different researchers conducted the administration of therapies to the experimental groups. The professionals responsible for applying the therapies in the different centers were Occupational Therapists and Physiotherapists specialized and trained in the techniques used.”
Ln 160: "EG1 received TE..." do you mean MT?
Response: has been amended and corrected. Reference was made to MT.
ln 179: number of children and place of residence. How are these data relevant for the analysis?
Response: Thank you for your comment, non-essential information has been removed from the table.
Results: please always acknowledge "n" for each percentage; do not repeat in the tables data already presented in the text.
Response: thank you for your comment. Modifications have been made.
Ln 394: due to study limitations, isn´t it more accurate to use the term "suggests" instead of "demonstrated"?
Response: thank you for your comment the modification has been made to the text.
Ln 418: data availability refers to where the research data associated with the paper is available and not to informed consent.
Response: we have modified the statement this way: “Data Availability Statement: The data that support the findings of this study are available from the corresponding author, [S.A.-P], upon reasonable request.”
We hope we have now answered all your comments and we are looking forward to hearing from you again.
Thank you very much,
Jessica Fernández Solana, OT, PT

Reviewer 4 Report

Comments and Suggestions for Authors

This study is about efficacy of upper limb rehabilitation after stroke, using different methods. It is a randomized, controlled and single-blind clinical trial comparing mirror therapy and cognitive therapeutic exercise (both with task oriented training) to a control group. 

It is an interesting topic in the field, but I have some interrogations about the method and results.

- The method is really similar to this described in the trial registered on the number NCT04163666 (Record History | ver. 4: 2022-06-24 | NCT04163666 | ClinicalTrials.gov). Is it the same (or modified one due to COVID) ?

- in study design, it is difficult to understand were the inclusion are done. Four locations are described, but further in the script, only two hospitals are announced (line 149 : "both hospitals").There is no mention about duration of participant's enrollment. 

- in procedure, it seems that experimental groups have 20 hours more of rehabilitation than control group.  The spread of rehabilitation over time, location and health professional conducting the rehabilitation are not specified.  A timeline and a study flow chart should be included to better understanding. 

- in instruments paragraph, there is no mention about minimal clinically important difference for the Fugl-Meyer assessment of the upper extremity, that could be important to correctly interpreted the results of the study (doi: 10.2522/ptj.20110009 / doi: 10.1589/jpts.31.917)

- on results, a flow chart with inclusion, non inclusion, exclusion, and lost of follow would be appreciated. Table 1 is redundant with the text. Figures or table with raw results of FMA scores at each assessment could be useful.

It seems that improvement is better in experimental groups than in control group, at 2nd and 3rd evaluation but without difference between the two experimental group (except for range of motion at 2nd evaluation). But these are only statistics results, and we can't know if the results are clinically relevant.  ANCOVA analysis are difficult to understand, and redundancy in the table are confusing. These statistics could be in Annexes. 

- in discussion, it seems difficult to compared mirror therapy and cognitive therapeutic exercise to control group, because this two therapies are combined with task oriented training, that is also used as a therapy itself (for example :  Effects of task-oriented training on upper extremity functional performance in patients with sub-acute stroke : a randomized controlled trial, jpts-31-082.pdf (nih.gov)). Also, there is no discussion about duration of rehabilitation (20 hours more), and that could impact the results. 

The benefit of MT or CTE can only be measured by comparing these two experimental group. With disponible data, it seems that there is no difference between the two techniques.  

On limitation paragraph, rather than the sample size, it seems that are the inclusion criterias that could limit the generalization of the results. COVID time is not mentionned before in the text. 

- Conclusion is not supported by the results. ("demonstrated"). The conclusion have to be rewrite. 

And brief remarks:

- sample size (lines 149-153) should be in statistics section 

- paragraph (lines 154-158) should be in study design

Author Response

Mr. Jessica Fernández Solana
Department of health sciences
University of Burgos, Paseo Comendadores s/n.
Burgos, 09001, Spain
Tel. (+34) 947499108
Email: jfsolana@ubu.es
12-01-2024
Healthcare. Subject: Submissions Needing Revision
Dear editor.
Thank you very much for inviting us to submit our response to reviewers for our manuscript (healthcare-2797806) entitled: “Efficacy of a Rehabilitation Program Using Mirror Therapy and Cognitive Therapeutic Exercise on Upper Limb Functionality in Patients with Acute Stroke.”
We have checked our manuscript according to the Academic Editor, the reviewers’ comments and the Journal requirements. We have also responded to some comments from reviewers point by point.
We would be very grateful if you could consider our manuscript to be published in your journal.
Yours sincerely,
Jessica Fernández Solana, OT, PT
D. Response to Reviewer 4:
First of all, we would like to express our sincere gratitude for all comments and suggestions received from the Reviewer 4. This information has certainly enriched the text for its best understanding, thank you very much indeed. We have clarified the reviewer4’s questions. We have introduced the required changes both in our answers to the specific comments and in the final manuscript V2.
This study is about efficacy of upper limb rehabilitation after stroke, using different methods. It is a randomized, controlled and single-blind clinical trial comparing mirror therapy and cognitive therapeutic exercise (both with task oriented training) to a control group.
It is an interesting topic in the field, but I have some interrogations about the method and results.
- The method is really similar to this described in the trial registered on the number NCT04163666 (Record History | ver. 4: 2022-06-24 | NCT04163666 | ClinicalTrials.gov). Is it the same (or modified one due to COVID)?
Response: thank you for your comment. Initially the registered trial was planned, however, modifications had to be made at the beginning of the study due to COVID. This led us to stop the study for a while and to make modifications to the respect of the intervention site. We had to change from home-based interventions to hospital-based interventions.
- in study design, it is difficult to understand were the inclusion are done. Four locations are described, but further in the script, only two hospitals are announced (line 149 : "both hospitals").There is no mention about duration of participant's enrollment.
Response: the network has been modified.
- in procedure, it seems that experimental groups have 20 hours more of rehabilitation than control group. The spread of rehabilitation over time, location and health professional conducting the rehabilitation are not specified. A timeline and a study flow chart should be included to better understanding.
Response: the experimental groups have received 20 extra hours of treatment. All three groups have carried out their usual therapy as prescribed by the rehabilitation physician. However, these experimental groups have received an extra 20 therapy sessions in which MT and CTE combined with task-oriented training have been applied. The therapy has been carried out in the mentioned hospitals. These complementary therapies in the study were carried out by an occupational therapist or a physiotherapist. We have changed the text to better explanation (lines 175-179).
We have also added a study flowchart (Figure 1).
- in instruments paragraph, there is no mention about minimal clinically important difference for the Fugl-Meyer assessment of the upper extremity, that could be important to correctly interpreted the results of the study (doi: 10.2522/ptj.20110009 / doi: 10.1589/jpts.31.917)
Response: We have added the clinically important difference in instruments paragraph.
- on results, a flow chart with inclusion, non inclusion, exclusion, and lost of follow would be appreciated. Table 1 is redundant with the text. Figures or table with raw results of FMA scores at each assessment could be useful.
Response: thanks for your comment, redundant data in the text have been removed. FMA raw scores have been added to table 3 and 4.
It seems that improvement is better in experimental groups than in control group, at 2nd and 3rd evaluation but without difference between the two experimental group (except for range of motion at 2nd evaluation). But these are only statistics results, and we can't know if the results are clinically relevant. ANCOVA analysis are difficult to understand, and redundancy in the table are confusing. These statistics could be in Annexes.
Response: Thank you for this comment; it is very useful for us. As you say, information is redundant in text and tables, so we have modified it. We removed and amended data form the text and we offer ANOCOVA results in tables as they seem to us more visual and better organized this way.
- in discussion, it seems difficult to compared mirror therapy and cognitive therapeutic exercise to control group, because this two therapies are combined with task oriented training, that is also used as a therapy itself (for example : Effects of task-oriented training on upper extremity functional performance in patients with sub-acute stroke : a randomized controlled trial, jpts-31-082.pdf (nih.gov)). Also, there is no discussion about duration of rehabilitation (20 hours more), and that could impact the results.
The benefit of MT or CTE can only be measured by comparing these two experimental group. With disponible data, it seems that there is no difference between the two techniques.
Response: thank you for your comment. We have tried to add to the discussion some literature to discuss our results with those of other studies. All our groups received their usual therapy in their referral hospital, and our experimental groups received 20 extra sessions with the indicated therapies, between which no significant difference in their application was seen.
On limitation paragraph, rather than the sample size, it seems that are the inclusion criterias that could limit the generalization of the results. COVID time is not mentionned before in the text.
Response: thank you for your comment. More information on the limitations produced by COVID has been added to this section.
- Conclusion is not supported by the results. ("demonstrated"). The conclusion have to be rewrite.
Response: the wording of this paragraph has been amended.
And brief remarks:
- sample size (lines 149-153) should be in statistics section
- paragraph (lines 154-158) should be in study design
Response: thank you for your comment, modifications have been made.
We hope we have now answered all your comments and we are looking forward to hearing from you again.
Thank you very much,
Jessica Fernández Solana, OT, PT

Reviewer 5 Report

Comments and Suggestions for Authors

(Comments I made here are correlated to highlighted text in the uploaded file)

Rows 149-153 - sample size for what? This is a comparative study aiming to find differences between interventions, and not a study about stroke, as I see it. A sample of 81 patients for 3 subgroups with interventions that are likely to be very close in terms of efficacy is at least underpowered (most rehabilitation interventions do not lead to marked differences in my experience, even in the case of a very carefully selected study group). Please explain the paragraph and what was the sample sized for, and if not for a comparative study please eliminate that part/repurpose/explain it properly. This undersizing of the study groups (in the context of a comparative study of 2 different interventions versus none) must be commented in the "discussion" section.

In reality what was proved is that intensive rehabilitation (of any type) is better than standard rehabilitation (and this is, unfortunately, a well established fact). 

Lines 159-161 – probably MT and CTE?

Line 159 – over how many days? 2/day 5d/week? 1/day?

Lines 165-167 and 168-169 repeat the same thing. It is not clear which were the tests performed before, immediately after and at three months from the intervention;

Line 171 – maybe an explanation/reference for Epidat 42?

Line 198 – the MT “modes” reflect in reality different techniques of rehabilitation (active therapy with MT supplementation, pure MT + movement imagination, assisted therapy with MT), and separate studies were performed just to evaluate them. The issue here is how were these three “modes” used in the patients – was it all uniform (all patients did all approaches)? Or, if not, how were they chosen? Were there any differences in outcome? Except the case the three “modes” were used more or less equally (and I think that the authors should mention that in the text), I suggest leaving out the information that will not be used in the analysis, maybe add it as a shorter description of the MT procedure after lines 202-204. In comparison, section 2.4.2 explains the process clearly, without any room for comments.

Row 261 and Table 1 – many data is presented without further use – marital status and number of children (I assume there is some connection with post treatment and home support), dominant side, type of stroke. Since they do trigger my (unsatisfied) curiosity as they are presented now, I suggest either removing these information or add short comments (if analysis was performed, of course) stating the relevance of those information (a.e. “no differences were found in the evolution of dominant side stroke as compared to non dominant hemisphere strokes”; “no difference was found between patients with family support and patients without family support”)

Rows 264-264 and Table 2 – data is not structured and it was difficult to really understand where are “differences between treatment groups” and where are “differential scores between the first and second evaluation”. Table 2 provides the data in a rather crude/raw form and needs to be structured and clarified

Rows 280-282 and Table 3 – same comment as above regarding this table.

As a personal curiosity (that may find its place in “results” and then “discussion”) – were there any significant differences between test 2 and test 3? (differences between active groups and controls were signifficantly larger at the 3rd test as compared to the 2nd test?). And, if so, did the difference correlate with anything? A better improvement for the active interventions groups after the intervention has ended would be really interesting in terms of neuroplasticity and in terms of proving that an early efficient intervention worths more than the same intervention administered at a later moment. An after-effect of the intervention means much more than the immediate benefit.

Line 318 – “approximately 4 OR 5 months”? “on average 4.x months” or “on median 4.x months”, or, at least, “between 4 and 5 months”.

Line 333 – the study doesn’t prove improvement in the ADLs. This can be inferred from the improvement in motor performance and the other studies you quoted

Line 351 – please review the sentence – probably “activation” is missing

Line 370-372 – please explain in the context of the experiment and of the previous phrase.

Lines 374-375 – is there any difference compared to lines 365-367?

Line 383 – “experienced TIME before COVID”? then maybe “several others”?

Comments on the Quality of English Language

In some paragraphs English does not flow as easily as it could. There are at least a few places where ideas are repeated over different paragraphs or where phrases are incorrect. The text needs to be revised with more care

Author Response

Mr. Jessica Fernández Solana
Department of health sciences
University of Burgos, Paseo Comendadores s/n.
Burgos, 09001, Spain
Tel. (+34) 947499108
Email: jfsolana@ubu.es
12-01-2024
Healthcare. Subject: Submissions Needing Revision
Dear editor.
Thank you very much for inviting us to submit our response to reviewers for our manuscript (healthcare-2797806) entitled: “Efficacy of a Rehabilitation Program Using Mirror Therapy and Cognitive Therapeutic Exercise on Upper Limb Functionality in Patients with Acute Stroke”
We have checked our manuscript according to the Academic Editor, the reviewers’ comments and the Journal requirements. We have also responded to some comments from reviewers point by point).
We would be very grateful if you could consider our manuscript to be published in your journal.
Yours sincerely,
Jessica Fernández Solana, OT, PT
E. Response to Reviewer 5:
First of all, we would like to express our sincere gratitude for all comments and suggestions received from the Reviewer 5. This information has certainly enriched the text for its best understanding, thank you very much indeed. We have clarified the reviewer5’s questions. We have introduced the required changes both in our answers to the specific comments and in the final manuscript V2.
Comments I made here are correlated to highlighted text in the uploaded file)
Rows 149-153 - sample size for what? This is a comparative study aiming to find differences between interventions, and not a study about stroke, as I see it. A sample of 81 patients for 3 subgroups with interventions that are likely to be very close in terms of efficacy is at least underpowered (most rehabilitation interventions do not lead to marked differences in my experience, even in the case of a very carefully selected study group). Please explain the paragraph and what was the sample sized for, and if not for a comparative study please eliminate that part/repurpose/explain it properly. This undersizing of the study groups (in the context of a comparative study of 2 different interventions versus none) must be commented in the "discussion" section.
Response: Thank you for your comment. This study is a comparative study that aims to find differences between interventions carried out with techniques that already have scientific evidence. However, in this case, we wanted to carry out two experimental groups in which these techniques are applied in combination, that is, in one group we applied the combination of Mirror Therapy and task-oriented training and in another group the combination of CTE and task-oriented training, with the aim of finding out whether the combination of these techniques is more effective than not applying them, or with which combination better results are obtained. The sample size was also calculated before the study was carried out:
"To determine the sample size, a formula adjusted for finite populations was used, considering a known proportion of strokes in the population based on data from the National Institute of Statistics (INE) [4], and a margin of error of 1% was estimated. The conclusion was that the sample should be composed of 81 stroke patients."
In reality what was proved is that intensive rehabilitation (of any type) is better than standard rehabilitation (and this is, unfortunately, a well established fact).
Response: Thank you for your comment. The efficacy of these therapies is proven, however, this study wanted to test the efficacy of MT and CTE evidence-based therapies in combination with task-oriented training. Also, to observe which of these combined therapies had a greater effectiveness on these patients.
Lines 159-161 – probably MT and CTE?
Response: thank you for your comment, it was a mistake, we meant MT.
Line 159 – over how many days? 2/day 5d/week? 1/day?
Response: this information has been added.
Lines 165-167 and 168-169 repeat the same thing. It is not clear which were the tests performed before, immediately after and at three months from the intervention.
Response: thank you for your comment. Duplicate information has been removed. The tests performed at all three points in time were the same and are detailed in the instruments section.
Line 171 – maybe an explanation/reference for Epidat 42?
Response: We have explained it is a freely available software for epidemiological analysis, which, among other things, allows sample calculations to be made.
Line 198 – the MT “modes” reflect in reality different techniques of rehabilitation (active therapy with MT supplementation, pure MT + movement imagination, assisted therapy with MT), and separate studies were performed just to evaluate them. The issue here is how were these three “modes” used in the patients – was it all uniform (all patients did all approaches)? Or, if not, how were they chosen? Were there any differences in outcome? Except the case the three “modes” were used more or less equally (and I think that the authors should mention that in the text), I suggest leaving out the information that will not be used in the analysis, maybe add it as a shorter description of the MT procedure after lines 202-204. In comparison, section 2.4.2 explains the process clearly, without any room for comments.
Response: thank you for your comment. The same methodology is used for all patients in the application of the TM and CTE modes.
Row 261 and Table 1 – many data is presented without further use – marital status and number of children (I assume there is some connection with post treatment and home support), dominant side, type of stroke. Since they do trigger my (unsatisfied) curiosity as they are presented now, I suggest either removing these information or add short comments (if analysis was performed, of course) stating the relevance of those information (a.e. “no differences were found in the evolution of dominant side stroke as compared to non dominant hemisphere strokes”; “no difference was found between patients with family support and patients without family support”)
Response: Thank you for your comment, information not relevant to the study has been removed from the table.
Rows 264-264 and Table 2 – data is not structured and it was difficult to really understand where are “differences between treatment groups” and where are “differential scores between the first and second evaluation”. Table 2 provides the data in a rather crude/raw form and needs to be structured and clarified.
Rows 280-282 and Table 3 – same comment as above regarding this table.
Response: In tables 2 and 3, two columns with the FMA raw scores obtained in the first, second and third assessment have been added to structure and clarify the results.
As a personal curiosity (that may find its place in “results” and then “discussion”) – were there any significant differences between test 2 and test 3? (differences between active groups and controls were signifficantly larger at the 3rd test as compared to the 2nd test?). And, if so, did the difference correlate with anything? A better improvement for the active interventions groups after the intervention has ended would be really interesting in terms of neuroplasticity and in terms of proving that an early efficient intervention worths more than the same intervention administered at a later moment. An after-effect of the intervention means much more than the immediate benefit.
Response: We have done this comparison, and no differences were found, so we have added “Finally, no differences were observed between the differential scores of the control group with the two experimental groups or between the two experimental groups when comparing second and third evaluations” at the end of results section.
We have also added some about it to the discussion.
Line 318 – “approximately 4 OR 5 months”? “on average 4.x months” or “on median 4.x months”, or, at least, “between 4 and 5 months”.
Response: 5 months after the initial assessment.
Line 333 – the study doesn’t prove improvement in the ADLs. This can be inferred from the improvement in motor performance and the other studies you quoted.
Response: thank you for your comment, this has been amended.
Line 351 – please review the sentence – probably “activation” is missing.
Response: thank you for your comment, this information has been added.
Line 370-372 – please explain in the context of the experiment and of the previous phrase.
Response:
We have modified this paragraph as follows:
“Some research state that it is not solely about the type of intervention but also about the duration of the therapy on which the results obtained may depend. Studies suggest that with a higher dosage of intervention, there are greater functional improvements [55]. In our research, both EG received more dosage of intervention, so it is also a variable to have into account.”
And we have also added some info to limitations as we feel it may be necessary:
“In addition, the total dose of treatment received was higher in the experimental groups than in the control group because the therapies under study were implemented at higher doses than the therapies the patients had already been prescribed, so it is necessary to take the results into consideration within this fact.”
Lines 374-375 – is there any difference compared to lines 365-367?
Response: thank you for your comment, in these lines the wording is the same, however the difference between them is that in lines 365-367 reference is made to the results between the first and the second evaluation, whereas in lines 374-375 reference is made to the results between the first and the third evaluation.
Line 383 – “experienced TIME before COVID”? then maybe “several others”?
Response: What we mean by this is that some of the patients became infected with COVID while the study was underway, so treatment sessions had to be stopped until they tested negative for COVID. Other patients have not been infected and others passed before the study started.
In some paragraphs English does not flow as easily as it could. There are at least a few places where ideas are repeated over different paragraphs or where phrases are incorrect. The text needs to be revised with more care.
Response: Thank you for your comment, English has been revised throughout the manuscript.
We hope we have now answered all your comments and we are looking forward to hearing from you again.
Thank you very much,
Jessica Fernández Solana, OT, PT

Round 2

Reviewer 2 Report

Comments and Suggestions for Authors

I congratulate you for the improvement of your article, corresponding to your important study.

Comments and suggestions:

1) I think that in the abstract it should be clear what the standard treatment of the Control Group is. Therefore, I propose adding that phrase (or with the corresponding acronym: TOT):

They were randomly allocated into three groups: a control group only for task-oriented training, and two groups undergoing either MT or CTE, both combined with task-oriented training.

2) It is advisable adding a figure with the mirror exercises, to illustrate the explanation of the text.

3) In lines 308-310, it should be discussed in which sense MT is superior to CTE (p=0.019) in range of motion and pain domain. Significant results should be discussed one by one in the DISCUSSION section.

4) In the first paragraph of the CONCLUSIONS, it would be convenient to add on line 404: but not for the range of motion and pain domain.

Author Response

Mr. Jessica Fernández Solana

Department of health sciences

University of Burgos, Paseo Comendadores s/n.

Burgos, 09001, Spain

Tel. (+34) 947499108

Email: jfsolana@ubu.es

26-01-2024

Healthcare.  Subject: Submissions Needing Revision

Dear editor.

Thank you very much for inviting us to submit our response to reviewers for our manuscript (healthcare-2797806) entitled: “Efficacy of a Rehabilitation Program Using Mirror Therapy and Cognitive Therapeutic Exercise on Upper Limb Functionality in Patients with Acute Stroke”

We have checked our manuscript according to the Academic Editor, the reviewers’ comments and the Journal requirements. We have also responded to some comments from reviewers point by point).

We would be very grateful if you could consider our manuscript to be published in your journal.

Yours sincerely,

Jessica Fernández Solana, OT, PT

  1. Response to Reviewer 2:

First of all, we would like to express our sincere gratitude for all comments and suggestions received from the Reviewer 2. This information has certainly enriched the text for its best understanding, thank you very much indeed. We have clarified the reviewer2’s questions. We have introduced the required changes both in our answers to the specific comments and in the final manuscript V2.

I congratulate you for the improvement of your article, corresponding to your important study.

Comments and suggestions:

1) I think that in the abstract it should be clear what the standard treatment of the Control Group is. Therefore, I propose adding that phrase (or with the corresponding acronym: TOT):

They were randomly allocated into three groups: a control group only for task-oriented training, and two groups undergoing either MT or CTE, both combined with task-oriented training.

Response: thank you for your comment. We have made the appropriate changes in the abstract.

2) It is advisable adding a figure with the mirror exercises, to illustrate the explanation of the text.

 Response: thank you for your comment. Figure 2 has been added showing patient positioning for Mirror Therapy.

3) In lines 308-310, it should be discussed in which sense MT is superior to CTE (p=0.019) in range of motion and pain domain. Significant results should be discussed one by one in the DISCUSSION section.

Response: thank you for your comment. However, no existing studies have so far found similar or contradictory results on this particular premise.

4) In the first paragraph of the CONCLUSIONS, it would be convenient to add on line 404: but not for the range of motion and pain domain.

Response: thank you for your comment. Done.

We hope we have now answered all your comments and we are looking forward to hearing from you again.

Jessica Fernández Solana, OT, PT

Reviewer 4 Report

Comments and Suggestions for Authors

Dear authors,

Thanks for the modifications of the manuscript, that improve the understanding of future readers. 

The new method description is better. Could you had information in 2.1 about inclusion period. "the sample... a stroke" should be delete (line 137-138). In 2.2, could you precise that it is a single blinded clinical trial (line 147) ? Could you add information about patients localisation : is the trial performed in hospitalized or out-patients ? and precise where the usual therapy is done (in hospital?). Could you also had informations about duration and time of session in usual therapy (to better understand how percentage of rehabilitation time was added in the experimental group compared to usual therapy) ? Could you correct the description of FMA-UE : the total score of 66 points is only for the motor score, sensory and range of motion/pain have another scoring. This could help to better understand raw results. 

In Results section, thanks for adding a flowchart. Could you reference it in the text ? Could you adapt the flowchart : it seems that there is a mistake in control group data (n=45 after allocation, 1 lost of follow up and 40 for analysis). Could you correct line 262, data were collected from 128 patients (rather than 120) - or rewrite the sentence ? Table 2 can be released and residual data (affected side and type of stroke) add in the text. Thanks for the raw results, there are very helpful. But the presentation still hard to understand. I think that it would be easier if there were presented in a figure with 4 illustrations (functionality / motor domain / sensory domain / ROM & pain), each representing score evolution along time for the 3 groups. Scale should be adjusted for minimum and maximum of each score. 

In discussion method, paragraph (line 322-327)is redundant and could be deleted. Paragraph (line 372-374) is also redundant. Paragraph about duration of therapy (line 367-371) should be in limitation results, and should be augment if there is a great improvement due to the extra sessions. To note, pleased delete "it's" in line 367. Paragraph (line 372-374) is redundant. 

And some remarks :

- line 26 : ... and another three months later... should be more comprehensive than ... and another at 3 months...

- line 36 : you can delete "or stroke"

- line 161, 362, 375, 387, 396, 398 (x2)  : extra space between words

- line 184 : a "," is lacking after references

- line 234 : is it "therapist" rather than "patient" ?

- line 384 : ...Finally, particular...

Even if it is not the goal of the study, it could be interesting to determine patients with better response (post-hoc analysis), eventually considering the minimal clinically importance difference, for an another article. 

Author Response

Mr. Jessica Fernández Solana

Department of health sciences

University of Burgos, Paseo Comendadores s/n.

Burgos, 09001, Spain

Tel. (+34) 947499108

Email: jfsolana@ubu.es

25-01-2024

Healthcare.  Subject: Submissions Needing Revision

Dear editor.

Thank you very much for inviting us to submit our response to reviewers for our manuscript (healthcare-2797806) entitled: “Efficacy of a Rehabilitation Program Using Mirror Therapy and Cognitive Therapeutic Exercise on Upper Limb Functionality in Patients with Acute Stroke.”

We have checked our manuscript according to the Academic Editor, the reviewers’ comments and the Journal requirements. We have also responded to some comments from reviewers point by point.

We would be very grateful if you could consider our manuscript to be published in your journal.

Yours sincerely,

Jessica Fernández Solana, OT, PT

  1. Response to Reviewer 4:

First of all, we would like to express our sincere gratitude for all comments and suggestions received from the Reviewer 4. This information has certainly enriched the text for its best understanding, thank you very much indeed. We have clarified the reviewer4’s questions. We have introduced the required changes both in our answers to the specific comments and in the final manuscript V2.

Thanks for the modifications of the manuscript, that improve the understanding of future readers.

The new method description is better. Could you had information in 2.1 about inclusion period. "the sample... a stroke" should be delete (line 137-138). In 2.2, could you precise that it is a single blinded clinical trial (line 147) ? Could you add information about patients localisation : is the trial performed in hospitalized or out-patients ? and precise where the usual therapy is done (in hospital?). Could you also had informations about duration and time of session in usual therapy (to better understand how percentage of rehabilitation time was added in the experimental group compared to usual therapy) ? Could you correct the description of FMA-UE : the total score of 66 points is only for the motor score, sensory and range of motion/pain have another scoring. This could help to better understand raw results.

Response: thank you for your comment. The sentence has been removed. However, I cannot provide specific dates for the inclusion period because, as it was a consecutive sampling, the inclusion of patients in the study took place throughout the study, and as patients were admitted we assessed whether or not they were suitable for the study.

The patients were already outpatients at the time we attended to them and they went to their hospital for rehabilitation (lines 168-169). However, there is no exact time that we can give on their usual therapy, this is decided by each rehabilitation doctor depending on the patient and the sequelae...

The FMA score has been correct (see lines 200-202).

In Results section, thanks for adding a flowchart. Could you reference it in the text ? Could you adapt the flowchart : it seems that there is a mistake in control group data (n=45 after allocation, 1 lost of follow up and 40 for analysis). Could you correct line 262, data were collected from 128 patients (rather than 120) - or rewrite the sentence ? Table 2 can be released and residual data (affected side and type of stroke) add in the text. Thanks for the raw results, there are very helpful. But the presentation still hard to understand. I think that it would be easier if there were presented in a figure with 4 illustrations (functionality / motor domain / sensory domain / ROM & pain), each representing score evolution along time for the 3 groups. Scale should be adjusted for minimum and maximum of each score.

Response: thank you for your comment. The figure reference has been added in the text (line 181). The erroneous sentence in the text has been removed, table 2 has been removed and the data has been included in the text. A representative figure of the raw results of the three groups in the three evaluations has been added at the end of the results section.

In discussion method, paragraph (line 322-327) is redundant and could be deleted. Paragraph (line 372-374) is also redundant. Paragraph about duration of therapy (line 367-371) should be in limitation results, and should be augment if there is a great improvement due to the extra sessions. To note, pleased delete "it's" in line 367. Paragraph (line 372-374) is redundant.

Response: thank you for your comment. We have made changes in the text

And some remarks :

- line 26 : ... and another three months later... should be more comprehensive than ... and another at 3 months...

- line 36 : you can delete "or stroke"

- line 161, 362, 375, 387, 396, 398 (x2)  : extra space between words

- line 184 : a "," is lacking after references

- line 234 : is it "therapist" rather than "patient" ?

- line 384 : ...Finally, particular...

Response: thank you for your comment. We have made the appropriate changes, except for the last, which we do not know exactly which paragraph you are referring to.

Even if it is not the goal of the study, it could be interesting to determine patients with better response (post-hoc analysis), eventually considering the minimal clinically importance difference, for an another article.

Response: As you state, it was not the goal of this study. We are pleased that you consider we could could develop another article determining determine patients with better response (post-hoc analysis), and the minimal clinically importance difference. We had not thought about it, but we will surely do so. Thank you very much for your comment, which encourages us to keep moving forward.

We hope we have now answered all your comments and we are looking forward to hearing from you again.

Jessica Fernández Solana, OT, PT
